# Occupational Health: Does Compliance with Physical Activity Recommendations Have a Preventive Effect on Musculoskeletal Symptoms in Computer Workers? A Cross-Sectional Study

**DOI:** 10.3390/ijerph18147604

**Published:** 2021-07-16

**Authors:** Sara Moreira, Maria Salomé Ferreira, Maria Begoña Criado, Jorge Machado, Cristina Mesquita, Sofia Lopes, Paula Clara Santos

**Affiliations:** 1ICBAS—Instituto de Ciências Biomédicas Abel Salazar, Universidade do Porto, 4050-313 Porto, Portugal; jmachado@icbas.up.pt; 2ESS|IPVC—Escola Superior de Saúde, Instituto Politécnico de Viana do Castelo, 4900-314 Viana do Castelo, Portugal; salomeferreira@ess.ipvc.pt; 3CBSin—Center of BioSciences in Integrative Health, 4000-105 Porto, Portugal; mbegona.criado@ipsn.cespu.pt; 4UICISA: E—Health Sciences Research Unit: Nursing, Nursing School of Coimbra (ESEnfC), Portugal School of Health, Polytechnic Institute of Viana do Castelo, 4900-314 Viana do Castelo, Portugal; 5IINFACTS—Institute of Research and Advance Formation in Health Sciences and Technology, 4585-116 Paredes, Portugal; 6CESPU—Departamento de Tecnologias de Diagnóstico e Terapêutica, Escola Superior de Saúde do Vale do Sousa, Instituto Politécnico de Saúde do Norte (IPSN), 4585-116 Paredes, Portugal; srl@ess.ipp.pt; 7LABIOMEP, Laboratório de Biomecânica do Porto, Universidade do Porto, 4200-450 Porto, Portugal; 8ESS|PPorto—Departamento de Fisioterapia, Escola Superior de Saúde, Politécnico do Porto, 4200-072 Porto, Portugal; ctmesquita@ess.ipp.pt (C.M.); paulaclara@ess.ipp.pt (P.C.S.); 9CIR—Centro de Investigação e Reabilitação, ESS|P, 4200-072 Porto, Portugal; 10CIAFEL—Centro de Investigação em Atividade Física, Saúde e Lazer, Faculdade de Desporto, Universidade do Porto, 4200-450 Porto, Portugal

**Keywords:** occupational health, musculoskeletal symptoms, physical activity level, computer workers, workplace

## Abstract

A lack of physical activity in computer workers (CW) can contribute to the development of musculoskeletal symptoms (MSS). **Aim:** (i) Evaluate MSS over a 12 month and 7 days period, (ii) determine physical activity (PA) levels and compliance with World Health Organization (WHO) PA recommendations, and (iii) assess the relationship between compliance with WHO PA recommendations and MSS. **Methods:** A cross-sectional observational study comprising 119 computer workers. The Nordic Musculoskeletal Questionnaire (NMQ) was used to evaluate the MSS and the International Physical Activity Questionnaire (IPAQ) was used to analyse the perception of the level of PA. **Results:** MSS occurred in the lumbar region (56.3%), neck (43.7%), and shoulders (39.5%). 44.7% of the participants reported a low level of PA. The percentage of compliance was similar among women and men (62.2% vs. 58.5%, respectively). Negative correlations were found between pain intensity and metabolic equivalent values. The participants who followed the WHO PA recommendations reported a lower frequency of MSS compared to those who did not, but the differences were not statistically significant. **Conclusion:** It was possible to conclude that computer workers presented a higher frequency of MSS in the lower back, neck, and shoulders. Regarding the level of PA, participants were mostly classified as low. Participants who followed the WHO PA recommendations reported lower MSS than those who did not. This finding could be important in obtaining successful programs that promote health-oriented physical activity in this group of workers.

## 1. Introduction

When analysing the literature on occupational health, it is possible to observe that concern about workers′ health has been growing [1]. In this sense, increasing physical activity (PA), besides the improvement of metabolism and cognition aspects and the reduction of stress, also seems to play a key role in reducing musculoskeletal symptoms (MSS) and thus reducing the economic and social costs associated with them [2]. In 2011, there was an estimated $213 billion in annual cost of direct treatment for and lost wages due to musculoskeletal disease in the United States [3].

Work-related musculoskeletal injuries (WRMI) are a “set of somatic disorders developed as a result of the cumulative action of repeated trauma and muscular strain of occupational activity” manifesting through musculoskeletal symptoms [4,5].

Depending on the profession, MSS occur in a variety of locations directly related to the function performed and the associated technical gesture. Physical factors, such as the overloading of some muscle groups, stress, prolonged maintenance of incorrect postures, repeatability of the same pattern of movement, and mechanical compression of body structures, are included in the risk factors for WRMI [4]. Other important factors that can influence the appearance of these injuries include: (i) individual factors, such as poor physical fitness, age (older people are at higher risk), and obesity; (ii) environmental factors, such as environments with inadequate lighting and temperatures, noise, and work environments that are not adapted to the needs of the worker, and (iii) organisational and psychosocial factors, such as monotonous tasks and low job satisfaction [6]. When a worker is constantly exposed to risk factors, the symptoms that were initially intermittent become permanent, interfering with both the ability to work and with daily life activities [4,5]. Work-related MSS represent a substantial disease burden and healthcare cost. In terms of the burden of illness, the Global Burden of Disease (GBD) study stated that MSS disorders were responsible for more than 120 million years lived with disability (YLDs) and accounted for more than 21% of disability worldwide [7].

Computer work (CW) has taken on a very important role in the daily life of many companies [4,5]. The type of tasks performed by CW lead to an increased sedentary behaviour, which lowers the physical activity level, and a lower postural variability [8,9,10]. These factors lead to an increased prevalence of MSS in these workers [11,12,13]. According to the study of Eltayeb et al. (2007) [14], in computer workers, the regions most affected by MSS include the cervical spine and shoulders, corresponding to an annual prevalence of 33 and 31%, respectively. In the hand, arm, elbow, forearm, and wrist there is an annual prevalence of 11, 12, 6% and 7%, respectively.

Over the past decade, increasing the physical activity level has become a public health concern that led to the publication of national and international recommendations.

According to the new World Health Organization guidelines, (WHO) 2020, in order to obtain health benefits, all adults between 18 and 64 years old must perform regular physical activities [15]. The previous stipulation that PA should be accumulated in at least 10 min [16] bouts has been removed [17,18]. At present, PA of any bout duration is associated with improved health outcomes. These new recommendations for adults specify a target range of 150–300 min of moderate-intensity physical activity and 75–150 min of vigorous-intensity physical activity per week [15].

In spite of the known importance of PA in the maintenance of health, the literature about the degree of adherence to the current recommendations among groups of populations is scarce and inconsistent findings were obtained. In the study of Alkerwi et al. (2015), different factors, such as education level, self-health perception, and the subjects’ awareness of the importance of exercise, were found as factors associated with the adherence to PA recommendations [19]. With respect to CW, Weenas et al. (2019) referred that white-collar worker had lower odds of complying to the WHO guidelines than other types of workers, but the effect of the occupation was not consistent throughout different models [20]. In this context, it is indispensable to investigate the perception of PA that those workers have and how to change attitudes in order to obtain successful programs that promote health-oriented physical activity in different social groups [21,22].

As far as is known, there are no studies about the relation between the compliance of PA recommendations and the frequency of musculoskeletal symptoms in computer workers in Portugal, but this kind of study would enable more efficient preventive and interventional approaches.

Thus, the aims of the present study were to: (i) evaluate MSS over a 12 month and seven days period, (ii) determine PA levels and compliance with WHO PA recommendations, and (iii) assess the relationship between compliance with WHO PA recommendations and MSS.

As such, the study hypothesis was that compliance with WHO recommendations leads to a decrease in the frequency of musculoskeletal symptoms in computer workers.

## 2. Materials and Methods

### 2.1. Study Design and Ethics

A cross-sectional observational analytical study was performed. Data were collected from CW adults. Ethical approval for this study was obtained from the Institute of BioMedical Sciences Abel Salazar Ethics Committee CHUP/ICBAS (963), and informed consent was given on the first page of the online questionnaire. All the participants were informed about the aims and procedures of the study, and they provided consent for their participation. The participants could refuse to participate in the study at any moment according to Law 67/98 of 26 October 1998 and the World Medical Association Declaration of Helsinki Ethical Principles for Medical Research.

### 2.2. Sample Recruitment

The target population of this study consisted of 424 CW from the automotive market company *BorgWarner* in Viana do Castelo, Portugal. The study focused on CW, who are a group who work approximately two thirds of their working hours in a sitting position and are therefore at increased risk for developing chronic illnesses due to their sedentary behavior. This sedentary work also implies less adequate postures, thus increasing the prevalence of WRMI [23,24]. The defined inclusion criteria were to be aged between 18 and 65 years old. The exclusion criteria included diagnosis of non-work-related medical conditions, such as ankylosing spondylitis, chronic joint diseases, neurological diseases, relevant (osteoarticular) surgeries, significant artificial joint replacement, articulation, multiple sclerosis, myotonic dystrophy, or neurodegenerative diseases and congenital malformations of the musculoskeletal system [25].

Before conducting the study, a visit to the company was made in order to make initial contact with the possible participants and their work environment.

The study questionnaires were entered into the Google Forms platform to make them faster and more accessible to complete. The questionnaire link was provided to the potential participants via email by the company’s human resources. The forms could be filled in from 5 to 12 June 2019.

The final sample consisted of 119 volunteers, 82 male and 37 female, aged between 21 and 50.

One participant was excluded for presenting a diagnosis of ankylosing spondylitis.

### 2.3. Questionnaires

The final questionnaire consisted of three parts: sociodemographic, Nordic Musculoskeletal questionnaire [26], and International Physical Activity Questionnaire—short version [27]. A pilot study was carried out in order to test any procedures. The survey was designed to take no longer than 15 min to complete and was a self-administered questionnaire.

#### 2.3.1. Sociodemographic Questionnaire

This questionnaire was designed by the main researcher in order to characterise the sample and collect sociodemographic data [28]. Firstly, general information, including sex, birth date, relational status, and education, was included. Anthropometric variables, such as weight and height, were self-reported, followed by the participants’ medical history and lifestyle. Finally, issues related to work were included (number of working hours per day, rotating or fixed shift, years of service, other paid employment in addition to the current one, and if so, how many hours in this extra activity per week, current employment status, type of employment contract, and managerial functions). To capture additional contextual information, number of hours per week of working in domestic and leisure activities were included.

#### 2.3.2. Nordic Musculoskeletal Questionnaire (NMQ)

The NMQ was validated for the Portuguese population and facilitated the evaluation of musculoskeletal symptoms [26]. It consisted of questions related to nine anatomical regions (neck, shoulders, elbows, wrist, thoracic region, lumbar region, hips, knees, ankles, and feet). The individual was asked if he/she had any symptoms in the last 12 months and in the last 7 days in one or more regions, and if in the last year he/she had to avoid his/her normal day to day activities due to symptoms. The anatomical regions were highlighted on a body chart so that there was no doubt about the area they were referring to [29]. The questionnaire also presented a numerical pain scale (NPS) that allowed the study participant to classify their pain in the “last 7 days” according to the indicated regions.

This questionnaire had moderate criterion validity and good reliability [26].

#### 2.3.3. International Physical Activity Questionnaire (IPAQ)—Short Version

The physical activity level was measured using the short version of the International Physical Activity Questionnaire (IPAQ), a questionnaire that had been shown to have acceptable reliability and validity for the Portuguese population [27]. It measured the frequency, duration, and intensity of physical activity over the previous 7 days. Each of these activities had an allocated “metabolic equivalent of a task” (MET) score [30]. To calculate the score, an Excel document developed by Dr. Hoi Lun Cheng of open access [31] was used. A cut-off point of 150 min per week of PA was used to classify the subjects to meet recommendations.

### 2.4. Statistics

The sample of 119 participants allows it to be possible to detect a medium effect size (h = 0.051) [32] in the chi-square test for the independence of MSS in relation to compliance with WHO recommendations for PA, with a power of 0.80 at the 0.05 significance level [33].

The data obtained from the questionnaires were processed in Excel for cleaning and normalisation purposes, then moved to be inserted into the IBM program SPSS Statistics Version 25.0 (*Statistical Package for the Social Sciences*^®^, IBM Corp Armonk, NY, USA) for posterior statistical analysis. This made it possible to make descriptive and inferential statistics considering a significance level of 5%. Descriptive statistics were used to characterise the sample through measures of central tendency, such as the mean, and dispersion measures, such as the standard deviation. Absolute frequencies and relative frequencies were used to characterise the sample.

To test the study hypotheses, inferential statistics were used through the application of the Student′s t-test for independent samples (between gender and vigorous, moderate, walking, and total METs), and the chi-square test of independence was used to verify the independence of categorical variables, along with *Spearman* (r_s_) correlations to ascertain whether there was a relationship between METs and pain intensity [34,35]. The following thresholds were considered to classify the strength of the correlations: |0.00–0.19| very weak, |0.20–0.39| weak, |0.40–0.59| moderate, |0.60–0.79| strong, and |0.80–1.0| very strong.

## 3. Results

### 3.1. Sample Characterisation

As can be seen in Table 1, the sample presented an average age of 34.3 years and a body mass index (BMI) of 25.06 kg/m^2^. Significant differences were found between genders in relation to BMI, with males showing higher values compared to females (25.8 vs. 23.1 kg/m^2^, respectively).

28.0% of the total sample were also found to be in pre-obesity, 50.8% were married or cohabiting, and 57.1% had a college degree, with females having the highest academic qualifications (*p* = 0.018), as shown in Table 2.

### 3.2. Musculoskeletal Symptomatology

The highest frequency of MSS in the last 12 months was in the lumbar region (56.3%), the neck (43.7%), and the shoulders (39.5%), with the three regions most frequently affected in both genders as shown in Table 3. However, there were statistically significant differences in the frequency of MSS between genders, with women having a higher frequency than men in the neck region (62.2% vs. 35.4%, *p* = 0.011), shoulders (54.1% vs. 32.9%, *p* = 0.048), thoracic region (24.3% vs. 8.5%, *p* = 0.038), and ankles and feet (45.9% vs. 22.0%, *p* = 0.015), as shown in Table 4.

When asked about avoiding their normal activities in the last 12 months, 56.3% of the women reported to avoid normal activities due to problems in the neck region.

When analysing the distribution of MSS by region in the last 7 days, it was found that the whole sample had the highest prevalence in the lumbar region (23.5%), followed by the neck (17.6%), and shoulders (12.6%), as shown in Table 3. Between genders, females showed greater evidence of neck symptomatology (32.4%) while males reported a predominance in the lumbar region (24.4%), as shown in Table 4.

Regarding pain intensity by region in the last 7 days, it was found to be slight for the whole sample, with the lumbar region presenting the highest pain intensity on average (2.2) and the ankles and feet the lowest (0.7).

There were no statistically significant gender differences in pain intensity by region except for the neck region, where women had a higher pain intensity than men (2.5 vs. 1.4, respectively, *p* = 0.022).

### 3.3. Physical Activity

Most computer workers reported a low level of PA (44.7%). Females showed a higher percentage of moderate PA levels (34.3%) when compared to males (24.1%), but this factor is reversed with respect to the high PA level, with males presenting a percentage of 31.6% and females 20.0%, as shown in Table 5. However, the variables are independent, so the level of PA does not depend on gender (*p* = 0.362).

When comparing the means of METs minute/week by gender, it was observed that males had a mean of vigorous METs minutes/week and a total of METs minutes/week that were both higher than females (*p* = 0.024, *p* = 0.004 respectively).

When analysing the average time spent by each gender per day of the week sitting, females showed more sedentary activity than males (*p* = 0.04). These results are shown in Table 5.

In assessing compliance with WHO PA recommendations, 59.7% of participants followed the recommendations. The percentage of compliance was similar among women and men (62.2% vs. 58.5%, respectively, *p* = 0.864) (Figure 1).

Correlations were made between METs and pain intensity. Significant low intensity negative correlations were found between vigorous METs and neck pain intensity (r_s_ = −0.329, *p* = 0.003), vigorous METs and low back pain intensity (r_s_ = −0.238, *p* = 0.031), and total METs and neck pain intensity (r_s_ = −0.249, *p* = 0.027) in males. In females, there was a negative correlation of significant weak intensity between moderate METs and wrist pain intensity (r_s_ = −0.340, *p* = 0.043).

With respect to the relationship between the MSS in the last 12 months and the MSS in the last 7 days, and compliance/non-compliance with the PA recommendations by gender, it was observed that the MSS variables by region and compliance or non-compliance with the recommendations were not statistically associated. These results are shown in Table 6 and Table 7.

MSS—musculoskeletal symptomatology; *n*—absolute frequency; %—relative frequency; independence chi-square test

## 4. Discussion

This study evaluated the frequency of musculoskeletal symptoms in the last 12 months and in the last 7 days in CW, determined the perception of PA level as well as the compliance with PA recommendations made by WHO, and verified if there was a relationship between compliance with the recommendations and the MSS. The musculoskeletal symptoms by regions most affected in the last 12 months and the last 7 days were similar, with the lumbar region being the most affected, followed by the neck and shoulders. However, as expected, the frequency was higher in the last 12 months, showing more than half of the symptoms in the lower back. With respect to the last 7 days, a frequency of neck, shoulder, and lumbar MSS in women was also revealed. The male gender showed a predominance of symptoms in the lower back, knees, and neck. These results are in agreement with the regions that are documented as having the highest symptoms in computer workers; neck, shoulders and lower back [14,36,37,38,39,40]. Various studies indicate that the presence of symptomatology in these regions is due to flexion postures adopted during long periods of work, associated with repetition of the task, causing greater tension in muscle and ligament structures [40,41,42].

Based on the results of the pain intensity, it was possible to verify a significant difference between genders. On the one hand, females presented higher levels of pain in the neck region. Some factors may contribute to these results. Firstly, according to literature, females have a higher prevalence of symptomatology, which is twice as high as in males [38,43]. This difference may be explained by the fact that when performing identical work tasks, women exhibit substantially greater muscle activity relative to their muscle capacity and strength [44,45]. Secondly, Portuguese women spend more time doing housework, as gender equality is not so well-established in Portugal, leading women to perform cumulative tasks and having less time for physical recovery and exercise [46]. A lack of recovery in leisure time, especially in terms of a lack of muscle relaxation, may increase the risk of disorders [44,45]. On the other hand, females presented an overall pain intensity reduction when compared with men. This may be related to the fact that the female participants were relatively young (females presented an average age of 33.3 years) and according to previous studies, age is one of the risk factors for the development of pain in computer users. Older workers are more likely to develop musculoskeletal symptoms [41,42].

The results of the IPAQ questionnaire made it possible to understand the perception of PA level between genders. Comparing genders, they are matched at low PA; however, females showed a higher level of moderate activity, while males showed a higher level of high/vigorous PA. These results agree with Nawrocka et al. (2017), who reported that women who do office work have a higher level of moderate and low PA compared to men [47]. Various studies reported that the male gender is more active compared to the female gender. However, this fact was not verified in the present study. The level of physical activity between the genders is identical, as it was registered that the gender variable and the level of physical activity were independent variables [48,49].

Regarding the METs values obtained, it was noticeable that males showed a significant difference in the average of vigorous and total METs compared with females. This can be explained by the male preference for vigorous activities [50]. However, these MET values showed a very high standard deviation, revealing a high value dispersion and heterogeneity of the sample in relation to the PA level.

As for sitting time per day, females showed a higher mean than males, with a significant difference. However, in contrast to these results, some studies showed that the total sitting time per day is significantly longer in men than in women [51].

In a first summary, the European Foundation for the Improvement of Living and Working Conditions indicated that women tend to carry out more repetitive work on average, while men are less likely to sit for prolonged periods in their work practice [14]. Excessive hours of sitting required by the profession, improper positions (lack of postural variability), repetitive manual tasks (these workers’ technical gestures involve using the keyboard and mouse, using repetitive movements), and low levels of physical activity, among other factors, have been associated with an increased risk of pain [41,42].

A negative correlation was found between neck pain intensity and vigorous and total METs, and low back pain intensity and vigorous METs in males. In females, this relationship occurred between the intensity of pain in the wrists and hands and moderate METs. This showed that the lower the metabolic expenditure, the greater the pain intensity in these regions. It is reported in the literature that low energy expenditure has short- and long-term detrimental effects on the health and physical capacity of computer workers [49,52]. Moderate and vigorous PA are associated with a decrease in C-reactive proteins and an increase in anti-inflammatory mediators, thus decreasing inflammation and, consequently, pain [47,53,54]. As such, physical exercise has protective effects and potential health benefits that will directly intervene in pain reduction [10,55,56]. PA also augments the neurogenesis capacity, producing greater emotional/psychological resilience and stability, giving better coping strategies to face life stress and adversities. Recent studies prove that it provides a state of well-being that enables individuals to realise their own potential [57].

With regards to PA recommendations, we found that almost half of the participants did not comply with the WHO PA recommendations, but males had a lower percentage compared to females. In the study by Nawrocka et al. [45], it can be observed that males had higher compliance with PA recommendations made by WHO than females, which is not corroborated by the results presented in the present study.

MSS were less frequent in the participants who followed PA recommendations, even though there were no statistically significant differences between the implementation of the recommendations and the MSS. It is well established that physical exercise improves musculoskeletal fitness, and there is growing evidence that increased musculoskeletal fitness is associated with an improvement in general health [58]. Several studies indicate that physical exercise is extremely important for the prevention of injuries and reduction of symptoms in computer workers [47,49,59]. In addition to physical exercise, factors such as age—older workers are prone to an increased risk of injury—and the work environment directly influence the appearance of these injuries [6,41,56,60,61].

As evidenced by a number of studies, good work environment conditions directly influence the prevention of MSS [6,8,41,42,56,60]. In this regard, the participants in the present study are young and the company has a good working environment: good lighting and low noise, adapted to the needs of each worker; it should be noted that these two factors are protective for the prevention of MSS.

One limitation of this study was the fact that the sample was formed of volunteers, producing a selection bias. Another limitation was that, out of a total of 424, only 119 computer workers answered the questionnaires, representing a non-response bias, as well as the fact that it was not possible to verify whether the workers answered the questionnaire themselves, as it was answered online.

The recruitment rate of 28.3% could have been improved through dissemination strategies in the company’s various internal communication channels [62]. Therefore, future studies should consider this approach. To address these limitations in future studies, it is recommended that a larger sample should be used, consisting of computer workers from several companies.

In addition, the population that was evaluated was formed of healthy adults; future studies should seek to include clinical populations.

## 5. Conclusions

The main aim of this study was to evaluate the relationship between PA and MSS.

It was possible to conclude that computer workers present a higher frequency of MSS in the lower back, neck, and shoulders. Regarding the level of PA, participants were mostly classified as low. Females showed a higher percentage of compliance with WHO PA recommendations when compared to males. These findings are important, as they can be used to adapt the promotion of PA levels.

Although no significant relationship was found between compliance with WHO PA recommendations and MSS in the last 12 months and the last 7 days, it was concluded that participants who followed the recommendations reported lower MSS than those who did not. In this context, further analysis is needed to confirm the results observed, but it seems clear that there is a necessity to direct efforts and resources in order to improve the adherence of computer workers to PA through the implementation of adequate programmes and policies, with the consequent social, economic, and environmental impacts on physically active populations.

## Figures and Tables

**Figure 1 ijerph-18-07604-f001:**
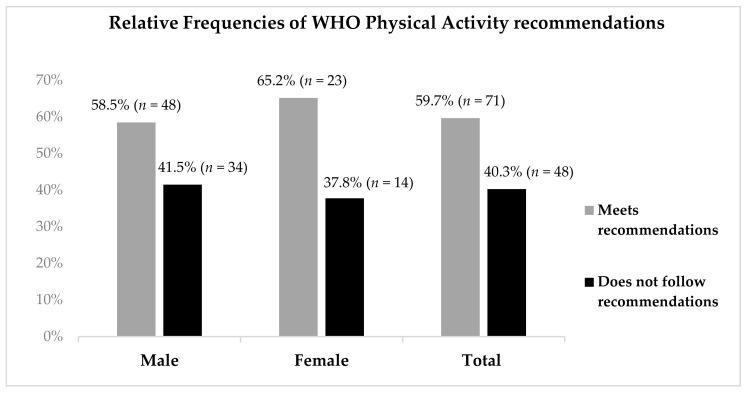
Relative frequencies of physical activity recommendations by gender (*p* = 0.864).

**Table 1 ijerph-18-07604-t001:** Characterisation of the sample according to age, height, weight, and BMI.

		Total	Female	Male	*p*
	*n*	Mean (SD)	Mean (SD)	Mean (SD)	
Age (years)	103	34.2 (6.88)	33.3 (6.71)	35.5 (6.97)	0.468
Height (m)	118	1.7 (0.09)	1.6 (0.05)	1.7 (0.06)	<0.001 *
Body mass (kg)	118	74.7 (15.02)	61.1 (9.61)	79.9 (13.56)	<0.001 *
BMI (kg/m^2^)	118	25.0 (4.06)	23.1 (3.23)	25.8 (4.25)	0.003 *

BMI—body mass index; SD—standard deviation, Student’s *t*-test * *p* < 0.05.

**Table 2 ijerph-18-07604-t002:** Characterisation of the sample by gender for BMI, marital status, and educational attainment.

	Total	Female	Male	*p*
	*n* (%)	*n* (%)	*n* (%)	
BMI Category				0.176
Low weight	3 (2.5)	2 (5.6)	1 (1.2)	
Normal weight	65 (55.1)	24 (66.7)	41 (50.0)	
Pre-obesity	33 (28.0)	7 (19.4)	26 (31.7)	
Obesity	17 (14.4)	3 (8.3)	14 (17.1)	
Marital Status				0.891
Single	53 (44.9)	16 (43.2)	37 (45.7)	
Married/Cohabitation	60 (50.8)	19 (51.4)	41 (50.6)	
Separated/Divorced	5 (4.2)	2 (5.4)	3 (3.7)	
Educational Attainment				0.018 *
Middle school	3 (2.5)	0 (0)	3 (3.7)	
High school	47 (39.5)	8 (21.6)	39 (47.6)	
School qualification	45 (37.8)	21 (56.8)	24 (29.3)	
Master′s degree	22 (18.5)	8 (21.6)	14 (17.1)	
Bachelor′s degree	1 (0.8)	0 (0)	1 (1.2)	
Professional course	1 (0.8)	0 (0)	1 (1.2)	
Total	119 ^(a)^ (100)	37 ^(a)^ (100)	82 (100)	

*n*—absolute frequency; %—relative frequency; BMI—Body Mass Index; independence chi-square * *p* < 0.05; ^(a)^ total participants in the BMI categorisation and marital status *n* = 118, total female participants in the BMI categorisation *n* = 36.

**Table 3 ijerph-18-07604-t003:** Musculoskeletal symptoms in different anatomical regions (*n* = 119).

	Musculoskeletal Symptoms			
Anatomical Regions	Symptomatology in the Last 12 Months	ADL Limitations in the Last 12 Months	Symptomatology in the Last 7 Days	Pain Intensity
	*n* (%)	*n* (%)	*n* (%)	Mean (SD)
Neck	52 (43.7)	16 (13.4)	21 (17.6)	1.7 (2.25)
Shoulders	47 (39.5)	11 (9.2)	15 (12.6)	1.4 (2.24)
Elbows	8 (6.7)	1 (0.8)	2 (1.7)	0.2 (0.87)
Wrists and hands	37 (31.1)	6 (5.0)	8 (6.7)	0.8 (1.63)
Thoracic	16 (13.4)	6 (5.0)	6 (5.0)	0.4 (1.44)
Low back	67 (56.3)	20 (16.8)	28 (23.5)	2.2 (2.79)
Hips and thighs	23 (19.3)	4 (3.4)	7 (5.9)	0.6 (1.51)
Knees	33 (27.7)	13 (10.9)	14 (11.8)	1.2 (2.20)
Ankles and feet	35 (29.4)	4 (3.4)	11 (9.2)	0.7 (1.49)

*n*—absolute frequency; %—relative frequency; ADL—activities of daily life.

**Table 4 ijerph-18-07604-t004:** Musculoskeletal symptoms, limitation in daily activities and pain intensity by gender (*n* = 119).

	Musculoskeletal Symptoms											
Anatomical Regions	Symptomatology Last 12 Months			ADL Limitation Last 12 Months			Symptomatology Last 7 Days			Pain Intensity		
	Male	Female		Male	Female		Male	Female		Male	Female	
	*n* (%)	*n* (%)	*p*	*n* (%)	*n* (%)	*p*	*n* (%)	*n* (%)	*p*	Mean (SD)	Mean (SD)	*p*
Neck	29 (35.4)	23 (62.2)	0.011 *	7 (8.5)	9 (24.3)	0.038 *	9(11.0)	12 (32.4)	0.010 *	1.4(1.91)	2.5(2.72)	0.022 *
Shoulders	27 (32.9)	20 (54.1)	0.048 *	6 (7.3)	5 (13.5)	0.314	7 (8.5)	8 (21.6)	0.071	1.2(2.02)	1.8(2.66)	0.211
Elbows	5 (6.1)	3 (8.1)	0.703	1 (1.2)	0 (0.0)	-	2 (2.4)	0	-	0.2(1.01)	0.1(0.39)	0.433
Wrists and hands	25 (30.5)	12 (32.4)	1.000	3 (3.7)	3 (8.1)	0.373	5 (6.1)	3(8.1)	0.703	0.9(1.79)	0.6(1.21)	0.391
Thoracic	7 (8.5)	9(24.3)	0.038 *	3 (3.7)	3 (8.1)	0.373	3 (3.7)	3(8.1)	0.373	0.4(1.47)	0.5(1.39)	0.755
Low back	46 (56.1)	21 (56.8)	1.000	12 (14.6)	8 (21.6)	0.497	20 (24.4)	8 (21.6)	0.923	2.3 (2.79)	2.1(2.83)	0.801
Hips and thighs	17 (20.7)	6 (16.2)	0.744	2 (2.4)	2 (5.4)	0.587	6 (7.3)	1(2.7)	0.433	0.5(1.31)	0.7(1.88)	0.630
Knees	23 (28.0)	10 (27.0)	1.000	8 (9.8)	5 (13.5)	0.539	10 (12.2)	4 (10.8)	1.000	1.3(2.26)	1.0(2.06)	0.422
Ankles and feet	18 (22.0)	17 (45.9)	0.015*	1 (1.2)	3 (8.1)	0.089	5 (6.1)	6 (16.2)	0.094	0.6(1.33)	1.1(1.74)	0.085

Fem—Female; n—absolute frequency; %—relative frequency; SD—standard deviation; ADL—activities of daily life; chi-square test of independence and Student’s *t*-test * *p*<0.05.

**Table 5 ijerph-18-07604-t005:** Absolute and relative PA level frequencies, values of vigorous, moderate, walking, and total METs and sitting time by gender.

	Total		Female		Male		
Physical Activity Level	*n* (%)		*n* (%)		*n* (%)		*p ^(a)^*
Low	51 (44.7)		16 (45.7)		35 (44.3)		0.362
Moderate	31 (27.2)		12 (34.3)		19 (24.1)		
High/Vigorous	32 (28.1)		7 (20.0)		25 (31.6)		
*METs*	*n*	Mean (SD)	*n*	Mean (SD)	*n*	Mean (SD)	*p ^(b)^*
Vigorous (minute/week)	119	782.1 (1244.11)	37	473.5 (728.50)	82	921.4 (1398.50)	0.024 *
Moderate (minute/week)	117	323.9 (685.74)	36	202.2 (253.38)	81	378.0 (802.58)	0.078
Walking (minute/week)	116	471.5 (671.49)	36	410.6 (445.99)	80	498.9 (752.20)	0.515
Total	*n*	Mean (SD)	*n*	Mean (SD)	*n*	Mean (SD)	*p ^(b)^*
Minute/week)	114	1595 (1867.14)	35	1093.5 (956.92)	79	1817.2 (2118.62)	0.013 *
Sitting time (hours/day)	107	5.9 (3.21)	32	7.2 (3.18)	75	5.3 (3.07)	0.004 *

SD—standard deviation; ^(a)^ p-value of independence chi-square test; ^(b)^
*p*-value of Student’s *t*-test; * *p* < 0.05.

**Table 6 ijerph-18-07604-t006:** Absolute and relative frequencies in the MSS in the last 12 months and PA recommendations by gender.

		WHO Compliance with Physical Activity Recommendations							
	Total			Male				Female	
MSS by Anatomical Region in the Last 12 Months	Yes	No	*p*	Yes	No	*p*	Yes	No	*p*
	*n* (%)	*n* (%)		*n* (%)	*n* (%)		*n* (%)	*n* (%)	
Neck	29 (40.8)	23 (47.9)	1.000	14 (29.2)	15 (44.1)	0.246	15 (65.2)	8 (57.1)	0.732
Shoulders	27 (38.0)	20 (41.7)	0.168	14 (29.2)	13 (38.2)	0.534	13 (56.5)	7 (50.0)	0.745
Elbows	4 (5.6)	4 (8.3)	0.161	3 (6.3)	2 (5.9)	1.000	1 (4.3)	2 (14.3)	0.544
Wrists and hands	20 (28.2)	17 (35.4)	0.713	14 (29.2)	11 (32.4)	0.810	6 (26.1)	6 (42.9)	0.470
Thoracic	8 (11.3)	8 (16.7)	0.219	3 (6.3)	4 (11.8)	0.441	5 (21.7)	4 (28.6)	0.705
Low back	37 (52.1)	30 (62.5)	0.595	25 (52.1)	21 (61.8)	0.519	12 (52.2)	9 (64.3)	0.515
Hips and thighs	13(18.3)	10 (20.8)	1.000	10 (20.8)	7 (20.6)	1.000	3 (13.0)	3 (21.4)	0.653
Knees	19 (26.8)	14 (29.2)	0.283	12 (25.0)	11 (32.4)	0.631	7 (30.4)	3 (21.4)	0.710
Ankles and feet	20 (28.2)	15 (31.3)	0.755	9 (18.8)	9(26.5)	0.575	11 (47.8)	6 (42.9)	1.000
Total	71 (100)	48 (100)		48 (100)	34 (100)		23 (100)	14 (100)	

MSS—musculoskeletal symptomatology; *n*—absolute frequency; %—relative frequency; independence chi-square test.

**Table 7 ijerph-18-07604-t007:** Absolute and relative frequencies in the MSS in the last 7 days and PA recommendations by gender.

	WHO Compliance with Physical Activity Recommendations								
	Total			Male			Female		
MSS by Anatomical Region in Last 7 Days	Yes	No	*p*	Yes	No	*p*	Yes	No	*p*
	*n* (%)	*n* (%)		*n* (%)	*n* (%)		*n* (%)	*n* (%)	
Neck	13 (18.3)	8 (16.7)	1.000	4 (8.3)	5 (14.7)	0.478	9 (39.1)	3 (21.4)	0.306
Shoulders	6 (8.5)	9 (18.8)	0.168	1 (2.1)	6 (17.6)	0.018*	5 (21.7)	3 (21.4)	1.000
Elbows	0 (0.0)	2 (4.2)	0.161	0 (0.0)	2 (5.9)	0.169	0 (0.0)	0 (0.0)	-
Wrists and hands	4 (5.6)	4 (8.3)	0.713	3 (6.3)	2 (5.9)	1.000	1 (4.3)	2 (14.3)	0.544
Thoracic	2 (2.8)	4 (8.3)	0.219	1 (2.1)	2 (5.9)	0.567	1 (4.3)	2 (14.3)	0.544
Low back	15 (21.1)	13 (27.1)	0.595	10 (20.8)	10 (29.4)	0.529	5 (21.7)	3 (21.4)	1.000
Hips and thighs	4 (5.6)	3 (6.3)	1.000	4 (8.3)	2 (5.9)	1.000	0 (0.0)	1 (7.1)	0.378
Knees	6 (8.5)	8 (16.7)	0.283	4 (8.3)	6 (17.6)	0.305	2 (8.7)	2 (14.3)	0.625
Ankles and feet	6 (8.5)	5 (10.4)	0.968	3 (6.3)	2 (5.9)	1.000	3 (13.0)	3 (21.4)	0.653
Total	71 (100)	48 (100)		48 (100)	34 (100)		23 (100)	14 (100)	

MSS—musculoskeletal symptomatology; *n*—absolute frequency; %—relative frequency; independence chi-square test * *p* < 0.05.

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
