# Peer review of "Occupational Health: Does Compliance with Physical Activity Recommendations Have a Preventive Effect on Musculoskeletal Symptoms in Computer Workers? A Cross-Sectional Study"

_ijerph, 2021, doi:10.3390/ijerph18147604_

Round 1

Reviewer 1 Report

Congratulations. The manuscript was improved, and I am recommending its acceptance.

Reviewer 2 Report

Dear authors,

I completely see the effort you put into the revised version of your manuscript.
However, I've some lingering comments, according the better readability to improve your manuscript's quality.

Abstract: Please thoroughly revise your Abstract according to minor grammar errors. I.e., punctuation (e.g., L30-34); sentence structure (e.g., L28).

Introduction: Revise the references, please. Make sure that the listing is in accordance with the occurrence in the manuscript. E.g., reference 2 occurs after reference 3.

L58-60: Please revise this sentence to provide the logical content of the statement. 

L78-83: I would prefer to place this content at the beginning of your Introduction.

Methods:

Please give clear references for the methods you applied in your study. Especially for the questionnaires at first mentioning; e.g. L138; L163/164.

L164: Remove the track changes from the manuscript (also L 267/268).

L188 et seq.: Please give references for the ranges of correlations.

Author Response

This manuscript is a resubmission of an earlier submission. The following is a list of the peer review reports and author responses from that submission.

Round 1

Reviewer 1 Report

General Comments

This manuscript aimed to evaluate the frequency of musculoskeletal symptoms in the last 12 months and in the last 7 days in computer workers and to analyze their association with physical activity level. The authors used the Nordic Musculoskeletal Questionnaire to assess the musculoskeletal symptoms and the International Physical Activity Questionnaire (IPAQ) to analyze the perception of the level of physical activity. Additionally, they also verified the compliance with WHO physical activity recommendations and checked whether there was a relationship between compliance with the recommendations and the musculoskeletal symptoms. The paper is well written and the approach is very interesting, but some important concerns should be addressed in order to improve the quality of manuscript.

Specific comments

1- The title should better reflect the objectives. There is no mention about “Educing in Labor Health” in the text. Moreover, the term “influence” brings the idea of a cause-effect relationship between musculoskeletal symptoms and physical activity level; however, it is difficult to establish this type of relationship considering the study design (best suited to an association analysis).

2- line 45: Does “back pain (13.9%)” refere to back pain in the thoracic region or back pain in general?

3- New World Health Organization (WHO) recommendations for physical activity were published in 2020. The reference should be updated. Also, the analysis of compliance with physical activity recommendations according to WHO should also be updated if necessary.

4- How were anthropometric variables (weight, height, and BMI) accessed? Were they self-reported? Please include this information in Methods section.

5- In the tables, I suggest to maintain the same order of the “groups” (Total/Male/Female, see table 2) to facilitate the interpretation of the results.

6- Table 3 needs formatting.

7- In Figure 2, there is a typo (“gende”). It also needs formatting.

8- Page 9, lines 281-289: The correlation coefficients should be included in the text.

9- Page 10, lines 309-310: The authors stated that they “found an overall pain intensity reduced in females. This is, perhaps, because the participants are relatively young”. However, this information does not seem to be fully supported by the results. Authors should comment on this.

10- Page 11, lines 391-394: The authors concluded that “although there seems to be no significant relationship between compliance to PA recommendations and MSS in the last 12 months and the last 7 days, it was observed that participants who followed the recommendations reported lower MSS than those who did not.” However, this information does not seem to be fully supported by the results. Authors should comment on this.

11- More than 50% of the references were published for more than five years ago (before 2015) and should be, at least in part, updated. Additionally, too much books were cited as references and it would be preferable if those references could be at least in part (and if relevant) replaced with articles.

Reviewer 2 Report

Congratulations. The subject of the study is excellent, but the manuscript must be improved. There are a lot of typos errors to be corrected. I am also suggesting adding some references to aid to improve the Introduction and Discussion sections. Moreover, I suggest including a hypothesis at the end of the Introduction section and a paragraph with the strength of the current work. All my suggestions are in the attached file.

Reviewer 3 Report

Thank you for the opportunity to review this manuscript. The purpose of the study was to investigate the musculoskeletal symptoms and actual physical activity level in computer-workers by standardized and validated questionnaires.
I see an acceptable interest of the topic for the readership of the scope of IJERPH. However, the manuscript needs major revisions before considered for potential publication in IJERPH.

Overall

  • The manuscript requires a revision by an institutional native speaker board (e.g., https://www.mdpi.com/authors/english).

Title

  • I am not quite sure if educing is the right word in this context.

Abstract

  • The Abstract is alright but needs improvement. Especially, the elaboration of the objective or purpose should be clearly outline and not mingled with other stuff, like the Methods.

Keywords

  • Reconsider your keywords, if they really represent the highlights and key of your study.
  • Do not use abbreviations as keywords.
  • I would assume you find more powerful keywords than sedentary lifestyle; additionally, you are rather analyzing working environment or working-related issues for human health.

Introduction

  • General 1: Try to really stick to the study relevant content in the Introduction.
  • General 2: I miss the clear research gap that you detected, the deduced research questions, and the associated hypotheses of your study.
  • L47-48: Sentence needs grammatical and content-related revision.
  • L48-52: Revise sentence according to length and grammar.
  • L72-73: According to this sentence, I’d guess you could re-structure your Introduction a bit neater. Try to be more conscious about breaking out general topic into specific content, leaving your research gap open, that you address with your specific research question, elaborated out of the logical structure of the Introduction.
  • L74-76: If you’re citing a direct citation, this requires mandatory referencing.
  • L76-78: Try to be really conscious about the real necessary content of your Introduction – I’d suggest, defining physical activity (PA) is not necessary in a journal entitled International Journal of Environmental Research and Public Health. Try to be more on point in your Introduction. Reduce unnecessary content.
  • L78-83: I’d guess the Canadian guidelines are developed based or aligned to the WHO guidelines. Therefore, it might useful to just report WHO guidelines. If not, WHO builds the framework for most national guidelines according to public and environmental health, therefore, it should need to come first.
  • L84-85: This is a vague statement. Enhance with giving us any numbers of change.
  • L88-90: With that sentence you conclude or kinda sum-up your Introduction; however, this is a bit sparse – you could elaborate that on a higher level – meaning elaboration of clear gap of knowledge, associated with a clear research questions that addresses the research gap – followed by the purpose of your study/analysis. Please, revise.
  • There is not enough reasoning with your theoretical elaborated content makes this analysis necessary. What is the bigger picture you want to address?
  • L93: If reporting names of institutions or similar, give always at first mentioning full affiliation in parentheses. I’d assume not lots people do know the company. So tell us exactly where it is.

Methods

  • Ethics communication should come after 2.1 Study Design. Because sample recruitment and everything else is based on your ethic approval.

Results

  • L214: I would put your sample characterization and the respective Tables into the Methods section. Or did you place deeper analysis / interpretation on the different groups and there specific characterization? If so, I’d assume you should motivate that. If you would dig into that, you should motivate, why to look into different gender groups; different education levels, and so on.
  • L244-246: Grammatically revise sentence and check thoroughly the punctuation throughout the manuscript – try to be consistent, also with punctuation of numbers!
  • L248-249: If you state that a result is higher or differs from another, I encourage you to report the inferential statistical metrics to underline the statement of difference!
  • Tables: Find more appropriate short titles for the Tables. The captions you deliver in the title are better fitting in the caption below the Table.
  • L259: If you consider a result as low, in relation to which other result?
  • L260-261: I’m a bit concerned now. Your just placing statements of differences by comparing two relative values in the results text sections, although you report all inferential stats in the Tables. However, with stating, again, that there is a difference on the relative values, you HAVE to report inferential stats also in the text. Importantly, because you provided all the required inferential stats in the Methods, however, your Results section is lacking on the inferential stats reporting.
  • L260-263: I do not see the necessity to interpret the reported results in that direction – and if so, it should be part of Discussion.
  • L267-269: That’s how I like to have differences reported; with at least reporting the p-value.
  • Please, be consistent with the decimal places, you’re reporting – decide consistently for two or three decimal places; additionally, check your manuscript, again, for the correct punctuation, especially of numbers; see for example Table 6, where you use dots or commas to report decimal values – stay to the American/anglo-saxon style, with dots as decimal separation and comma as separating 1000s.
  • L277-278: You cannot state such a strong interpretation (“better”) without having any inferential statistical support; if placing that statement, you need to tell that this is based on descriptive analysis.
  • Figure 2: You have to re-edit this figure, totally. It appears totally un-edited in my document and please use same font size and style for all your illustrations.
  • L281-284: That’s alright, how you report the results: however, based on that, you need to report in the Methods section, the ranges of r-values, which separate low from moderate and high correlations.

Discussion and Conclusion

  • General: I’d guess that your Discussion could clearly benefit by a clear separation of the findings in relation to the musculoskeletal symptoms and work and the correlation to physical activity level.
  • L299: At first, please, repeat the main purpose of your study before reporting the main finding.
  • L300: ”revealed a frequency”; this is a quite open and undirected statement; please be more straight!
  • L308: Please revise “About pain intensity”; maybe something similar to, “Based on the results of pain intensity…”
  • L308-310: Please, disentangle this sentence and the included statements for better understanding.
  • L310-311; This statement appears totally arbitrary. Why is this your assumption? Can you refer to any source, why you assume, age is a factor?
  • L311 et seq.: These associated findings, which support your findings could be better aligned and connected with each other; it appears a bit of unstructured composition of arguments, which help to support your findings, but it would be helpful to bring all the arguments together into the bigger picture here.
  • L319-322: Reduce sentence length for better understanding.
  • L324-328: Same here. Reduce sentence length; build shorter exclamations and assumptions to make it easier for the readership to follow your argumentation, especially throughout the Discussion.
  • L334-336: Don’t you have a explanation for your reverse results? Is there any potential bias considering the duties the women have to do in the company, which could also explain the results of more sedentary work in ladies?
  • L336 et seq.: This rather appears as a first conclusive statement; you should edit it respectively to be clearly separated by the before-hand argumentations. And maybe, it is also helpful, to introduce such statements with phrasing like “in a first summary…”.
  • L348-349: This happens occasionally. You report a finding of your study. Afterwards you communicate what other authors reported. However, there is often a missing link / association / connection between both arguments; or a missing explanation or interpretation of yours, guessing of your divergent results. This is the main part of the discussion: to clearly elaborate the own findings in relation to findings of other studies.
  • L363-367: Revise into at least two statements.
  • L369-371: Grammatically revise.
  • L373-375: Before-hand, you rather exclude the environmental conditions and their influences on the MSSs and PA…Yet, this is valid argumentation. Why not including these arguments earlier? And if this is valid, wasn’t there anybody who scientifically approved that?
  • I encourage you to briefly sum-up your discussion in one or two sentences before reporting limitations.
  • L383-384: Is that really the one and only way to easily overcome your limitations? I’d guess there is more to be done that just saying, recruit a larger sample from various companies. What about potential power analysis that could be done by your study for subsequent studies?
  • Also, I miss a clear outlook and/or practical implication in your conclusion.

Round 2

Reviewer 3 Report

Dear Authors,

Thanks for the in-depth and thorough work for enhancing your manuscript. I totally see the overall improvement. However, I’ve some lingering minor comments before the final consideration for publication.

Generally:

There are some comments left to be addressed. Are there any reasons you did not comment them or did the recommended revision accordingly?

Title: I agree that you improved the title, yet, I do not see that you really stated question, do you? Please revise grammatically!

Abstract:

L25-26: Please grammatically revise “…has brought about…”

L27: “Aim: Evaluate:” Please revise sentence for better readability. Repetitive occasion of colons is rather unusual.

Keywords:

No further comments.

Introduction:

L75-77: What is the contribution of the statement? Is that really fitting here? I would motivate to move such general phrases to the beginning of the Intro or to withdraw from the manuscript as it does not really contribute new information to the Intro. It’s rather appearing as a repetition of the arguments you raised already at the very beginning of the Intro.

L82-84: Please, thoroughly revise your manuscript for minor spelling and punctuation errors!

L92-94: Please grammatically revise your research question.

Methods/Results:

I still do not see the reasoning of putting the sample characterization as a first result into the Results section, as you do use the information only for sample description and you do not apply any analysis on the sample characteristics, don’t you? Therefore, I would have put that into the Methods. However, if you like it better that way, I can come along with that.

Tables:

I don’t think that you understood my comment on the Tables. Tables need a clear title, as “Table 3. – Musculoskeletal symptoms in different anatomical regions.” The rest of the information you also placed into the titles belongs to the captions under the Tables. Therein you explain what’s to be found in the Table and what the abbreviations mean. A three-line title of Table is not useful. Please revise!

Discussion/Conclusion:

You put valuable work into this section and I’m good with these parts.
